# communications
# earth & environment

# Protected areas reduce deforestation and degradation and enhance woody growth across African woodlands

Iain M. McNicol [1✉], Aidan Keane [1], Neil D. Burgess[2,3], Samuel J. Bowers[1], Edward T. A. Mitchard [1] & Casey M. Ryan [1]

Protected areas are increasingly promoted for their capacity to sequester carbon, alongside biodiversity benefits. However, we have limited understanding of whether they are effective at reducing deforestation and degradation, or promoting vegetation growth, and the impact that this has on changes to aboveground woody carbon stocks. Here we present a new satellite radar-based map of vegetation carbon change across southern Africa's woodlands and combine this with a matching approach to assess the effect of protected areas on carbon dynamics. We show that protection has a positive effect on aboveground carbon, with stocks increasing faster in protected areas (+0.53% per year) compared to comparable lands not under protection (+0.08% per year). The positive effect of protection reflects lower rates of deforestation (−39%) and degradation (−25%), as well as a greater prevalence of vegetation growth (+12%) inside protected lands. Areas under strict protection had similar outcomes to other types of protection after controlling for differences in location, with effect scores instead varying more by country, and the level of threat. These results highlight the potential for protected areas to sequester aboveground carbon, although we caution that in some areas this may have negative impacts on biodiversity, and human wellbeing.

[1] School of GeoSciences, University of Edinburgh, Edinburgh EH9 3FF, UK. [2] United Nations Environment Programme – World Conservation Monitoring Centre (UNEP-WCMC), Cambridge CB3 0DL, UK. [3] Centre for Macroecology, Evolution and Climate, Natural History Museum, University of Copenhagen, Copenhagen, Denmark. ✉email: i.m.mcnicol7@gmail.com

Protected areas, and other area-based conservation measures, remain a flagship strategy for reducing carbon emissions and biodiversity losses due to land use and land cover change[1,2]. Their importance was reflected in the Convention on Biological Diversity's Strategic Plan for Biodiversity (2011–2020), and in Target 3 of the recently agreed Kunming–Montreal Global Biodiversity Framework that sets out an ambition for 30% coverage of protected and conserved areas by 2030[3]. However, questions remain over the extent to which protected areas (hereafter PAs) are effective at meeting their multiple aims[4], leading to calls for evidence-based assessments of the performance of existing protected and conserved areas, to inform future expansion efforts[1].

Evidence regarding the effectiveness of PAs in delivering positive outcomes for nature and climate has increased rapidly over the past decade[5–9], although progress has been limited by our ability to accurately map changes in habitat structure across large areas. Indeed, whilst much work has focused on the ability of PAs to reduce deforestation, it has been harder to quantify if they can mitigate habitat degradation. These subtle changes in vegetation structure, which are typically caused by processes including overharvesting, inappropriate fire regimes, and other changes in land management, and can be important in driving carbon emissions[10–13], and lead to major changes in biodiversity[14,15]. There is similar lack of knowledge over the extent to which aboveground woody carbon (AGC) storage may be increasing via vegetation growth[16] and whether PAs have a role in mediating these patterns, e.g. by altering activities that suppress growth such as fire and herbivory. This information is particularly important with many countries likely to use PAs way of meeting international climate obligations. This may involve reducing carbon emissions from land use change, e.g. as part of Nationally Determined Commitments (NDCs) to the UNFCCC, as well as maintaining areas of vegetation growth in existing wooded areas, or enhancing growth through active restoration efforts e.g. as part of the Bonn Challenge[17].

The absence of such data on PA effectiveness reflects the considerable methodological and practical difficulties in measuring aboveground carbon storage, and especially its changes over time[18]. Studies reporting the impacts of PAs on deforestation, and degradation[7] have done so using optical satellite data, which used in isolation, is limited to measurements of the visible (2D) surface, and so provide no information on the vertical structure of the vegetation, and thus limited insights into its carbon density. Smaller changes in AGC, such as those related to vegetation growth, and lower intensity degradation, are therefore particularly difficult to measure using satellite remote sensing data, as they typically result in no clear changes to the tree canopy/ spectral signature (i.e. greenness) of the surface meaning optical measurements can miss a large proportion of these events[10,19].

High-resolution (tens of m) active remote sensing data, such as synthetic aperture radar (SAR), is known to be sensitive to 3D vegetation structure and, when combined with in situ measurements of tree biomass from inventory plots, can be used to accurately map changes in AGC stocks over large enough areas to allow an examination of the role PAs play in mediating these patterns[10]. This includes improved estimates of degradation[20–22], as well as insights into vegetation growth[10], which is often limited to sparse field inventory plots[16], or derived across large areas (i.e. tens of km) using coarse resolution satellite data[23] or models[24], none of which is suitable for assessing the impacts of PAs. A limitation of current spaceborne SAR systems is that when used in isolation their relationship with woody biomass saturates at relatively low AGC densities (<50–75 Mg C ha$^{-1}$), which precludes their application across moist tropical forests, but makes them useful in savanna, woodland and dry forests, which together are the largest land cover in the tropics and an important part of the global carbon cycle[25,26].

Here, we present new datasets showing changes in aboveground woody carbon (AGC), and associated changes in land cover, including deforestation, degradation and growth across the world's largest savanna ecoregion—the southern African woodlands—which cover 2.5 million km$^2$ over 7 countries (Fig. 1). We build upon our previous work[10] to map changes in AGC between 2007–2010 and 2015–2018 using the open access ALOS PALSAR and ALOS-2 PALSAR-2 annual mosaic product, created by the Japanese Space Agency (JAXA)[27]. The datasets are derived using a new approach designed to account for uncertainties in these mosaics, and reduce the soil/vegetation moisture-related error on the estimates of change. Using these change maps, we examine the extent to which PAs are effective at preserving AGC stocks, by promoting vegetation growth, and/or limiting anthropogenic change, i.e. deforestation (the loss of wooded area) and degradation (a reduction in carbon density of wooded areas), inside their boundaries. We also examine how PA effectiveness varies between countries, and PA types, separating areas under strict protection (such as national parks and IUCN category I–II areas) from other PAs, including forest reserves, and other areas principally managed for wildlife. To overcome the location bias by which PAs are often located in remote, less-used landscapes[28], coarsened exact matching is used to identify comparable protected and unprotected areas for analysis[29], based on covariates reflecting accessibility, extractable value and potential agricultural suitability.

The southern African woodlands are an important study region for evaluating PA impacts given their high biodiversity value, including many charismatic, but threatened animal and plant species which are endemic to the region[30]. African woodlands are unusual in that they retain significant wooded area and carbon stocks, alongside a large human population closely dependant on ecosystems for food, fuel, timber, and construction materials[31,32]. Increasing demand for resources is leading to widespread, and rapid deforestation and degradation, particularly around rapidly expanding urban centres and road networks[10,33–35], with regrowth in more remote areas thought to be counteracting these losses[10]. The time period covered by our dataset means PAs established after 2014 were excluded from the analysis, at which point they covered 21% of the region (7–35% of each country's land mass area; Fig. 1), with this large extent meaning they likely play a critical role in mediating carbon dynamics and patterns of human disturbance.

## Results

**Carbon and land-cover dynamics**. Region-wide AGC stocks were likely constant over the study period, estimated at 6.1 [95% CI: 5.4–6.7] Pg C in 2007–2010, and 6.2 [5.5–6.8] Pg C in the period 2015–2018 (Fig. 1), with net losses in Mozambique and Tanzania offset by net gains in the other countries (Fig. 2). The small net change in AGC (ΔAGC) obscures large gross losses from deforestation (0.2 [0.17–0.23] Pg C) and degradation (0.2 [0.17–0.23] Pg C), and gross gains from vegetation growth (0.6 [0.52–0.68] Pg C) (see 'Methods' for a definition of these land cover changes). The total wooded area, defined here as any 25 m pixel with an initial AGC density >10 Mg C ha$^{-1}$, was 2.6 [2.4–2.8] M km$^2$, with carbon gains detected in 55% [47– 61%] of this area, and deforestation and degradation occurring in 7.2% [6.0–9.1%] and 8.4% [7.0–10.0%] of the wooded area, respectively (Fig. 2 and Supplementary Data 1). The remainder of the wooded area (29.1% [23.5–36.0%]) experienced minor losses of AGC (<20%), which are likely caused by quasi-natural processes e.g. fire, herbivory[36].

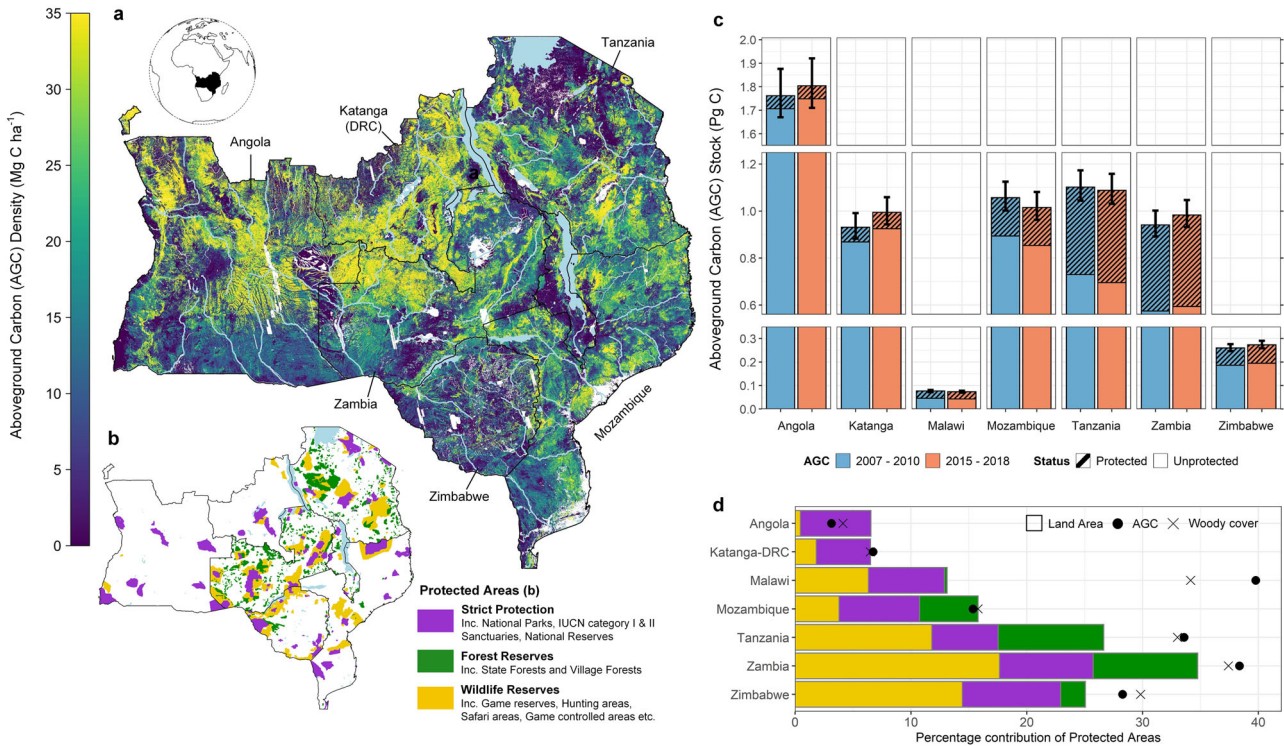

**Fig. 1 Spatial distribution of aboveground woody carbon stocks. a** The aboveground woody carbon (AGC) density across our study region for the period 2015–2018, and **b** the location of protected areas included in this study, separated by broad management category. **c** The total national-level AGC stocks calculated across wooded areas (>10 Mg C ha⁻¹) for the periods 2007–2010 and 2015–2018, including the 95% confidence intervals (CIs) that represent for the total each bar (i.e. protected and unprotected areas combined). **d** The percentage of each countries AGC, land area and wooded area that is contained inside protected areas.

PAs contain 21%, and 18% of the total wooded area and AGC stocks, respectively, although their importance varies by country, with PAs in Malawi and Tanzania containing a disproportionate amount of the national AGC stocks, relative to their areal extent (Fig. 1). PAs are not exempt from anthropogenic disturbance, with 21,000 [19– 25,000] km², or 4.1% [3.4–5.2%] of the PA being deforested, and a further 31,000 [28–42,000] km², or 6.2% [5.2–7.9%] undergoing a major loss (i.e. degradation) of AGC stocks (Fig. 2 and Supplementary Data 2). Smaller PAs appear more susceptible to these losses with 65% of PAs <100 km² in size exhibiting higher deforestation and/ or degradation rates than the region-wide rates reported above, compared to 32% of PAs >1000 km² (Fig. 2). Carbon gains were more prevalent inside PAs than outside, with 60% [52–66%] of protected wooded areas increasing in AGC. This, combined with lower deforestation and degradation rates, mean PAs make an outsized contribution to the overall carbon balance, with AGC stocks increasing at faster rate (+0.62% yr⁻¹ [0.52–0.71 % yr⁻¹]; total increase 0.54 Pg C ≡ 54 Million Tonnes (Tg) [49–60 Tg]) compared to unprotected areas (+0.12% yr⁻¹ [0.07–0.18% yr⁻¹]; 48 [30–67] Tg C).

**The PA effect on carbon stock change.** The differences in carbon and land use dynamics between protected and unprotected areas described above reflect, in part, a location bias, with PAs tending to be located in relatively inaccessible parts of the study region where anthropogenic disturbances are less likely (Supplementary Figs. 1 and 2). After matching, we find PAs have an overall positive effect on aboveground woody carbon (AGC), with stocks increasing by +0.53% yr⁻¹ [0.43–0.62% yr⁻¹] (2.79 [2.19–3.54] Tg C yr⁻¹) inside matched PAs, compared to +0.08% yr⁻¹ [−0.05–0.21% yr⁻¹] (0.40 [−0.23–1.12] Tg C yr⁻¹) in the

matched unprotected sample (Fig. 3). The small difference to the region-wide results presented in the previous section reflects the exclusion of the more remote and inaccessible parts of the PA network (45% of total protected wooded area) for which there are no suitable counterfactual areas to compare against (Supplementary Fig. 1). This is an important feature of our approach whereby PAs with no high quality matches are excluded since it is generally better to obtain an estimate of effectiveness with limited bias.

Overall, the carbon benefits of protection primarily reflects a reduction in carbon losses from deforestation, which were 42% lower in PAs compared to matched controls (1.26 [0.99–1.42] vs 2.15 [1.70–2.43] Tg C yr⁻¹, meaning that avoided deforestation accounts for 40% of the overall effect of PAs on ΔAGC (Figs. 3 and 4). We find PAs have a smaller but important effect on carbon losses via degradation (31% lower: 1.41 [1.17–1.60] vs 2.1 [1.71-2.33] Tg C yr⁻¹), and on carbon sequestered via vegetation growth (10% higher: 6.86 [5.74–7.85] vs. 6.15 [5.14–7.03] Tg C yr⁻¹), both of which contribute 28% to the net PA effect on ΔAGC, with the residual 4% due to a small reduction in minor losses. The biomass growth rate in areas that increased in AGC is similar in matched protected and unprotected areas averaging +0.55 ± 0.2 Mg C ha⁻¹ yr⁻¹ (mean ± SD), as is the intensity of both deforestation (−13.2 ± 6.0 Mg C ha⁻¹), and degradation (−8.0 ± 2.7 Mg C ha⁻¹), which may indicate similar types of agriculture and harvesting activities under protection. As such, the positive carbon outcomes in PAs largely reflects proportionate reductions in the areal extent of deforestation (−38% lower in matched PAs), degradation (−25%), and increases in the area of vegetation growth (+12%) (Supplementary Fig. 3). Our results are in keeping with previous assessments of avoided deforestation[6,37], and to the best of knowledge, directly show

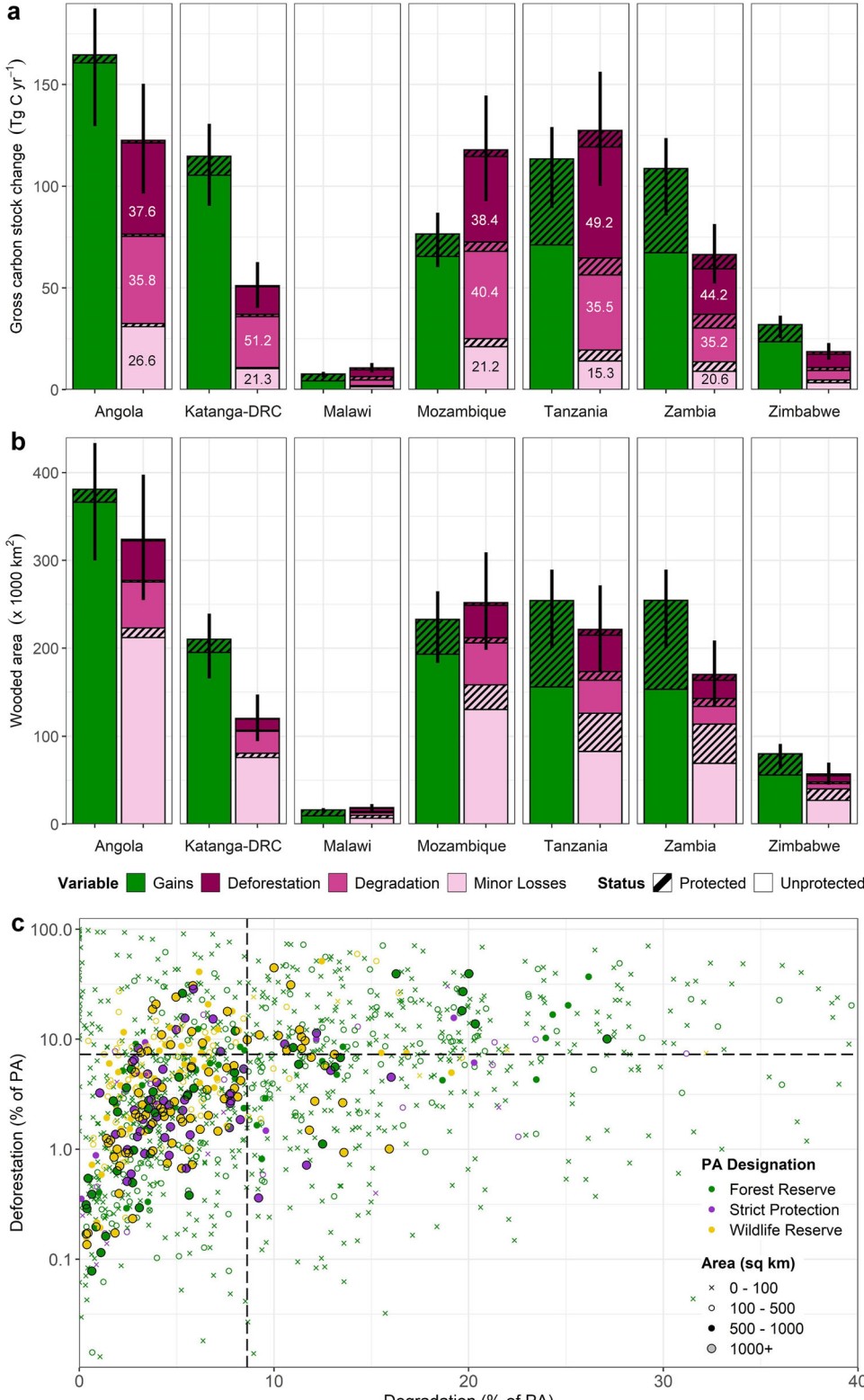

**Fig. 2 Gross carbon stock, and land cover changes across countries and protected areas. a** The area of deforestation, degradation, minor losses, and growth in each country, and **b** the total carbon stock changes resulting from these processes in Tg C yr$^{-1}$ (1 Tg = 1 million metric tonnes [Mg]). In (**a**), the values inside the section of each bar show the percentage contribution of each loss component to the total losses for that country. The vertical bars indicated the 95% confidence intervals (CIs) and represent for the combined uncertainty on each bar (i.e. total losses combined, as well as protected and unprotected areas). **c** The percentage of the initial wooded areas (as of 2007–2010) that was deforested and degraded for all individual PAs included in the analysis. The hatched lines show the percentage of the entire study region impacted by each process over the study period.

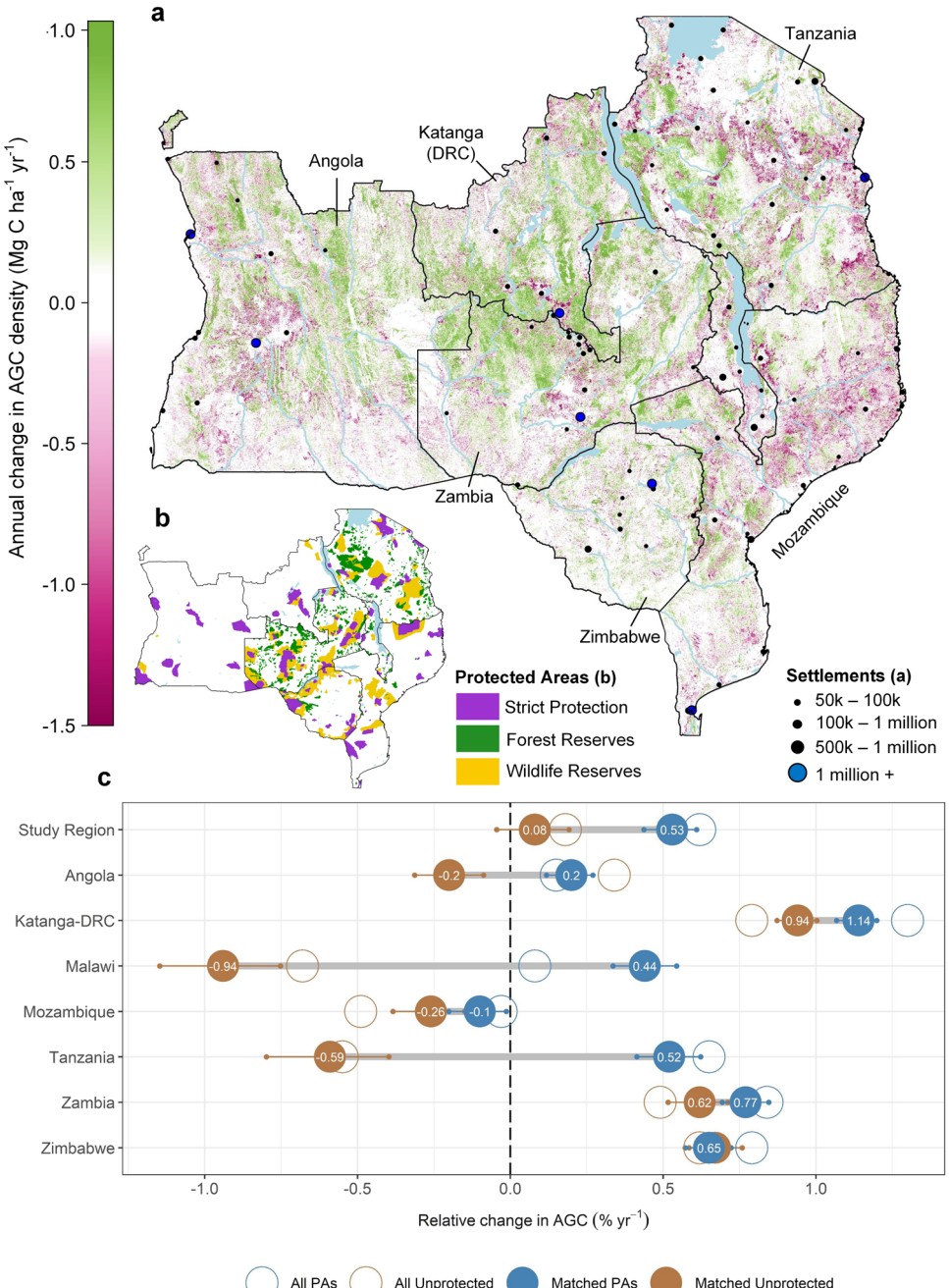

**Fig. 3 Change in aboveground woody carbon density across the southern African woodlands. a** The annual absolute change in AGC stocks (Mg C yr$^{-1}$) between the period 2007–2010, and 2015–2018. Changes are expressed on an annual basis as the average time-difference between the two maps varies across the study region. **b** The location of protected areas in 2014, separated by broad management type, is included again to ease comparisons between patterns of AGC change, and PA location. **c** The net relative change in AGC stocks over the study period across the subset of matched protected, and unprotected areas (solid circles), and across the entirety of the protected and unprotected areas in the region (unmatched; open circles). Here, changes are expressed relative to the initial AGC stock in each area, rather than in absolute terms, given the differing size of each country, and thus magnitude of each change. The 95% confidence intervals (CIs) around each point indicate the range of possible outcomes based on propagating the uncertainties in the model used to convert the radar data into AGC. The grey lines between solid circles highlight variations in PA effectiveness, i.e. the difference in outcomes between the matched protected, and unprotected areas. The results are broken down by country and management type in Supplementary Fig. 4.

for the first time that avoided degradation, and increased vegetation growth are important carbon benefits of protection.

The overall trends of PAs reducing land cover change, and increasing AGC stocks relative to their matched counterparts is consistent across countries (Figs. 3 and 4). However, there are large differences in the effect size of protection between national PA networks, with those in Tanzania, Malawi, and to a lesser extent Angola having the greatest matched effect on ΔAGC (Fig. 3). In contrast, the comparatively rapid gains in AGC observed across PAs in Katanga-DRC, and to some degree Zambia and Zimbabwe, are nearly equalled by gains in matched unprotected lands, meaning their PA networks are assessed as less effective at enhancing ΔAGC (Fig. 3) compared to unprotected areas.

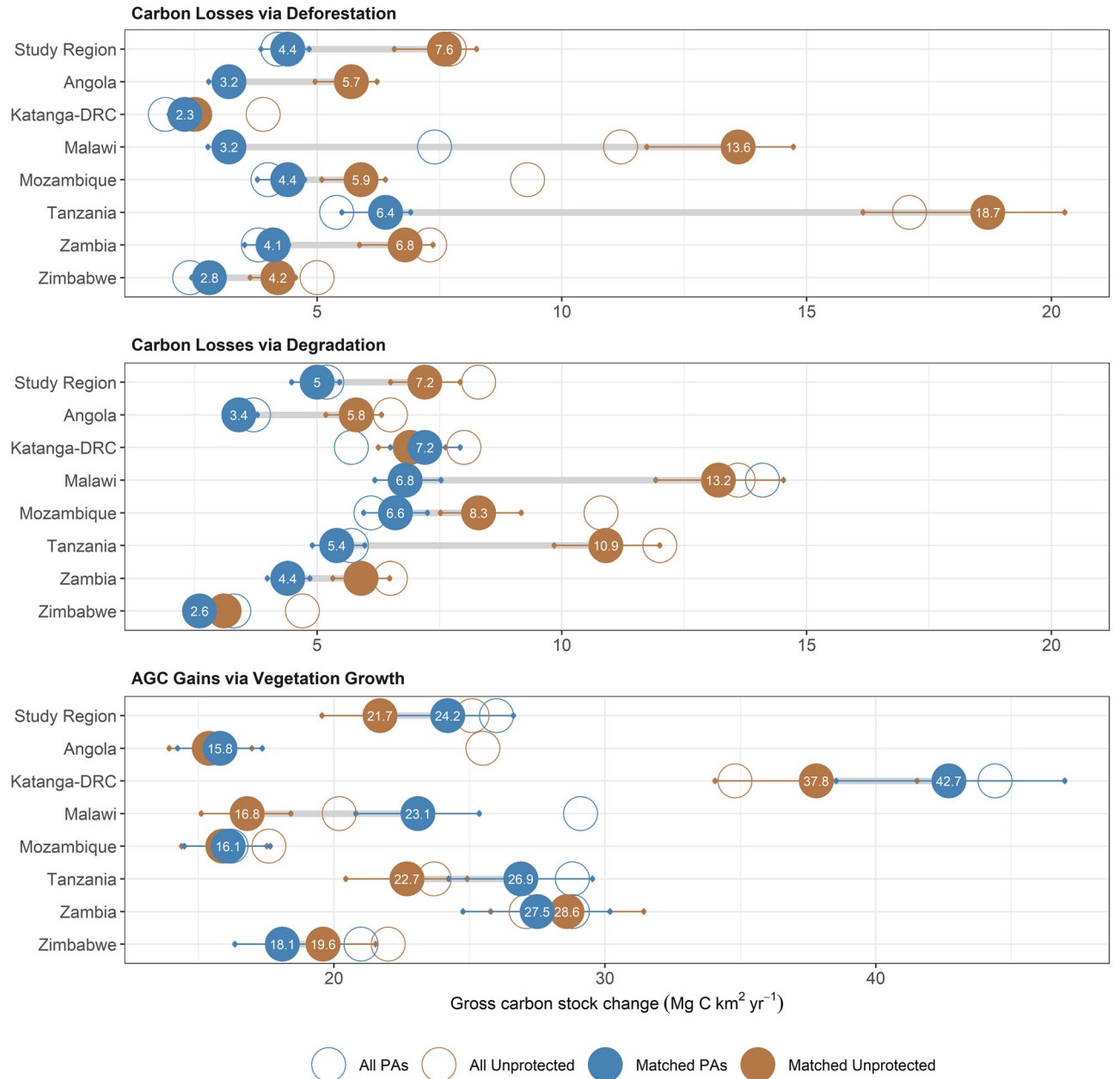

**Fig. 4 Comparison of the gross carbon stock changes due to deforestation, degradation and vegetation growth in protected and unprotected areas.**
Changes are expressed as a proportion of the total land area due the differing size of each country, and thus total magnitude of each change. The larger solid circles show the results for a matched subset of PAs, whilst open circles show the results for the full protected area in the region. The 95% confidence intervals (CIs) around the matched estimates indicate the lower and upper bounds on our estimates for both protected and unprotected areas based on propagating the uncertainties in the model used to convert the radar data into AGC. In this case, overlapping uncertainty bounds do not indicate the absence of an effect as a scenario that yields a lower rate of change inside a protected area would result in a lower rate inside the matched unprotected area.

**Variation in the effectiveness of PAs on carbon outcomes.** The broad national-level trends (Figs. 3 and 4) provide insights into the role PAs play in overall carbon balance, and their potential to contribute to national-level efforts to reduce carbon losses from deforestation and degradation. These results are naturally weighted towards larger, better matched PAs, and obscure the considerable within-country differences in effect scores related to PA designation (i.e. National Parks, Forest Reserves etc.), and heterogeneity between individual PAs (Supplementary Figs. 3–5 and Supplementary Data 3). Understanding the impact PA designation has on effect scores is particularly important given plans to expand the network and the financial commitment, and

potential social costs (e.g. restricted access to resources) associated with creating new PAs.

To that end, we compare carbon outcomes in areas likely to be under strict protection (36% of the total area protected; largely IUCN categories I–II, and National Parks), from those which are not, with these split if they are designated as Forest Reserves (21% of area), or managed principally as Wildlife Reserves (43%). The results show that overall, each of these groupings results in positive outcomes on ΔAGC, with the overall difference compared to matched unprotected areas estimated at +0.45% yr$^{-1}$ for strict protection, +0.60% yr$^{-1}$ for forest reserves, and +0.33% yr$^{-1}$ for wildlife reserves, although

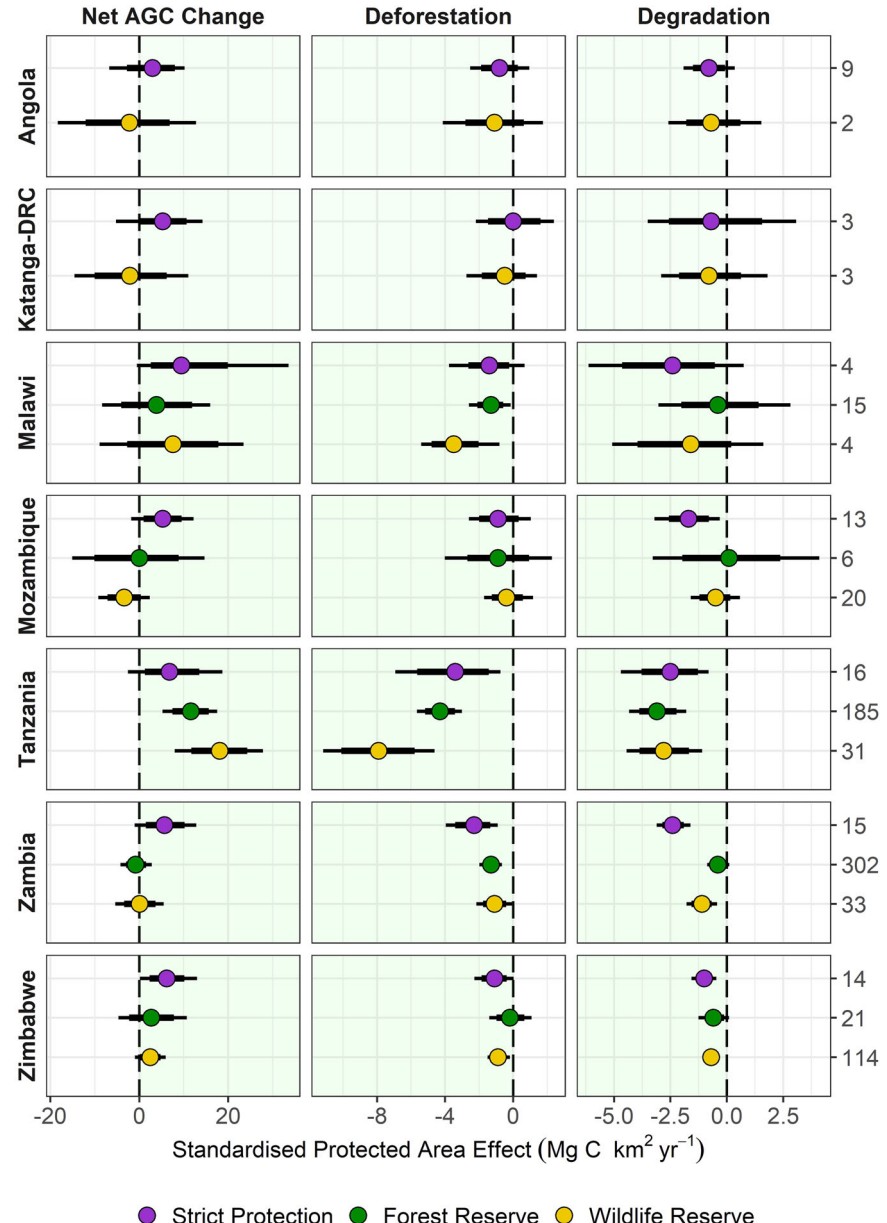

**Fig. 5 Comparison of the standardised PA effect on carbon stock changes between management types, controlling for differences in location.** Bayesian regression modelling is used generate an estimated PA effect for each country/management type by setting all other covariates to their mean i.e. the effect an average PA, accounting for all other observed sources of variation (see 'Methods'). Circles indicate the mean difference-in-difference in outcomes between protected and unprotected areas and their associated 95% Credible Intervals (CIs) (thin segment of error bar) and 80% CIs (thick). We interpret estimates whose 80% CIs do not overlap zero as a statistically meaningful effect. The areas shaded in green show where PAs reduced land cover change or enhanced ΔAGC compared to matched, unprotected areas, i.e., positive values mean matched PAs have better AGC outcomes (Mg C ha$^{-1}$ yr$^{-1}$), while for deforestation and degradation, a negative value indicates lower carbon losses.

the pattern, and size of the effect varies somewhat by country (Supplementary Fig. 4). As with protected and unprotected areas, direct comparisons between PA types are confounded by biases in location; for example, a large number of forest reserves tend to be located in areas with high land use intensity (Fig. 2), thus providing a greater opportunity for these PAs to have a larger effect. To account for this, we calculated standardised estimates of the PA effect on ΔAGC, and the gross carbon losses from deforestation and degradation (Fig. 5) using a Bayesian regression procedure which accounts for observable differences related to accessibility, extractable value and potential agricultural suitability, as well as country and management status (see 'Methods').

After removing the confounding effects of location, we find no clear evidence that PA designation has an impact on the effect scores with PAs under stricter protection generally showing slightly stronger effects, although the effect is not meaningfully better than both forest or wildlife reserves (standardised difference on ΔAGC < 0.1 Mg C ha$^{-1}$ yr$^{-1}$). There are exceptions to these patterns, most notably in Tanzania where both wildlife and forest reserves have a larger positive effect on ΔAGC compared to strictly PAs.

PAs with higher initial AGC densities were associated with more positive outcomes on ΔAGC, and larger reductions in deforestation and degradation, relative to matched unprotected areas (Supplementary Fig. 6). This may reflect differences in

management effectiveness, e.g. whether more effort is placed on protecting higher biomass PAs, or differences in the threat level in the matched controls, e.g. whether lower AGC areas are at less risk of exploitation meaning PAs have less opportunity to be effective. Areas with a higher suitability for cash crops[38], and those further from major cities, a proxy for urban demand for resources, were also associated with larger effects of PAs on ΔAGC. For cash crops, this reflects reductions in deforestation relative to their matched counterparts, however, increasing distance to cities has no impact on the effectiveness of PAs at avoiding deforestation. Instead, the positive impact reflects reductions in degradation, indicating that underlying drivers of these losses (e.g. demand for timber and charcoal) are less limited by distance, compared to deforestation (see ref. [12]). The size of the PA had no clear impact on its effect on ΔAGC, although larger PAs are associated with lower deforestation.

## Discussion

We find PAs are, on average, associated with increases of aboveground woody carbon (AGC), above that in matched unprotected areas. This is the result of avoided deforestation and degradation and associated carbon losses, and also an increase in the areal extent of vegetation growth, relative to non-PAs. These three processes contribute 39%, 28% and 28% of the net carbon benefits of protection, respectively. The relatively even split across the three key land cover processes is important as it highlights that previous analyses which solely analysed deforestation effects are missing many of the carbon benefits of PAs[39]. Our results also reaffirm known patterns of AGC change across our study region[10], with carbon losses concentrated around urban centres and road networks, and carbon gains in more remote areas with sparse population, and/or a relatively high (>1000 mm yr$^{-1}$) mean annual precipitation (e.g. northern Angola, western Zambia, southern DRC).

Here, we show that PAs also play an important role in shaping these patterns, which for the first time is shown to include the prevalence of vegetation growth, which is 12% higher compared to matched unprotected areas, albeit with no differences in the rate at which carbon is accumulating (0.55 Mg C ha yr$^{-1}$). These carbon gains encompass both the growth of mature vegetation and wooded areas re-growing following disturbance, but do not include growth in non-wooded areas (<10 Mg C ha$^{-1}$), or non-wooded to wooded transitions, due to additional uncertainties on change detection in low AGC areas. Whilst these carbon benefits may help address the climate change mitigation agenda, it is important to acknowledge that increasing woody biomass in more open ecosystems can be associated with a loss of biodiversity[40], and in parts of our study region, may directly conflict with biodiversity management goals. There is likely a dichotomy between the wetter regions of the study (the Miombo woodlands) where increasing woody cover is rarely perceived as a biodiversity threat, and associated with higher faunal and floral diversity[41,42], and the more open arid ecosystems where it is a key concern of land managers[43]. As a result, the finding of more extensive woody AGC increases in PAs might result in a mixed picture for biodiversity conservation in some parts of the region.

Degradation, which in this carbon-focused context refers to a reduction in the carbon density of the vegetation, without a transition to very low woody biomass often associated with a new land cover (i.e. deforestation), has previously been shown to be a major source of carbon loss from these ecosystems[10,44]. In this study, we find that degradation accounted for 49% of the combined anthropogenic (i.e. deforestation and degradation) carbon losses, confirming its importance as a major source of land use change emissions, albeit one not easily accounted for in the global

carbon budget[45]. We anticipated that PAs would be less effective at reducing degradation compared to deforestation, as the transient activities that lead to the degradation of AGC, such harvesting for timber and charcoal[34], would be harder to reduce in PAs than the more sedentary activities that cause deforestation, which is often driven by agricultural expansion[36]. People travel large distances to collect potentially valuable resources, with PAs potentially containing more tree species of harvestable size for timber and charcoal due to historically lower rates of disturbance, meaning even inaccessible areas may be targeted for timber extraction[12,46,47]. Our findings support this hypothesis, with matched PAs showing a smaller reduction in degradation rates (−25%) and associated carbon emissions (−31%), compared to deforestation (−38% and −41% respectively).

This general pattern is broadly consistent, although it obscures considerable heterogeneity in effect scores between countries (Figs. 2 and 3), and among individual PAs (Supplementary Fig. 5). At the national-level, our results show there is relatively little difference between countries in terms of the deforestation and degradations rates inside PAs. Instead, the variations in effect scores largely reflects the varying pressures PAs are under, specifically, the rate of loss in matched unprotected areas, and whether they are located in areas suitable for growing cash crops, and/or in close proximity to major cities.

Each of our broad PA designations were associated with positive outcomes on carbon storage, although we find no evidence that this has a meaningful impact on effect scores, once biases in location are accounted for. Indeed, those PAs more likely to place greater restrictions on land use, i.e. those with IUCN management category I-II, and/or National Parks, result in similar outcomes to PAs less likely to limit human use, and/or receive less support than more prominent conservation areas[48]. Tanzania is a notable exception, with forest reserves and wildlife areas outperforming strictly PAs. The reasons for this are unclear, especially the outsized impact of forest reserves, which likely receive less funding that strictly PAs[48]. The comparatively large impact of wildlife reserves may reflect, in part, the positive influence of Selous Game Reserve, which constitutes 17% of the Tanzanian dataset. It is designated at IUCN category IV area and so not considered strictly protected, although it is a UNESCO World Heritage Site, and since 2019, part of it has been reclassified as a National Park.

Interpreting the effects of PAs is somewhat limited by the lack of detailed information on the management system that is in place (e.g. IUCN category, METT score), and the governance strategy (e.g. community vs centrally managed) for the majority of the PAs in our study. Such data would permit a more nuanced analysis as to how these factors influence PA effectiveness[7,8] and is a target for future research. That being said, our results do suggest that stricter, or more intensive conservation is not a prerequisite for preventing deforestation and degradation, and demonstrates the potential for well-managed, inclusive, PAs contribute to national-level efforts to reduce emissions from deforestation and degradation, alongside their other potential benefits. This conclusion has potentially broad implications, as more restrictive conservation practices are more likely to result in negative social outcomes[49,50], by limiting or even stopping access to resources, and displacing local communities.

It is equally as important to note that for many PAs, though perhaps more so for wildlife reserves, the finding of positive carbon outcomes is unlikely a core target of protection, nor in some PAs, is such an outcome necessarily desirable if it conflicts with efforts to protect faunal diversity. For example, elephants, and other browsing herbivores are known to be a limiting factor to AGC[14,51], meaning our findings of positive carbon outcomes may indicate negative trends in animal densities. However, we

consider it more likely that this will impact on the prevalence of minor losses, and on the suppression vegetation growth, with observed reductions in deforestation and degradation more likely to indicate reductions in human activity[36], which may be a positive outcome[26].

The carbon benefits of PAs relative to unprotected areas also need to be placed in context; firstly, it is possible that some of the benefits of protection are undermined by displacement (i.e. leakage) of livelihood activities to other areas[52], rather than a real reduction in e.g. deforestation and degradation. Secondly, it should also be noted that despite their effectiveness at reducing anthropogenic losses, PAs are not immune to widespread disturbance, with deforestation and degradation rates found to be very high in some PAs, and associated carbon losses from PAs totalling 6 Tg C yr$^{-1}$ (60 million tonnes), equivalent to 15% of the region-wide annual emissions from these processes. We also caution against direct comparisons to our previous estimates of AGC change for the region given the different approaches used to derive these values, and the decadal time frame covered in this study. The remote sensing data is also averaged over multiple years to overcome inconsistencies in the annual mosaics, and to account for potential biases associated with choosing a single pair of years for analysis. As such, some short lived (1-2 years) change events may be missed, which is more likely to affect estimates of degradation (due to regrowth, and because it is often a precursor to deforestation). However, the advantages of the longer time frame is that it leads to higher signal to noise ratios for detecting land cover change. As such, we can be more confident that the observed changes are real and that our findings accurately reflect the impact of PAs on the regional carbon balance.

## Methods

**Mapping carbon and land use change.** The datasets were broadly created using methods described in reference[10], which presented maps of AGC densities and change for the period 2007–2010 at 25 m resolution, using a combination of satellite radar images, specifically the ALOS PALSAR mosaic product (Ver. 1), produced by the Japanese Space Agency (JAXA)[27], and in situ carbon stock estimates for converting radar backscatter to estimates of AGC. Change detection is based on a probabilistic approach that takes into account uncertainties in the radar data itself, and the conversion of the radar backscatter to aboveground carbon densities. A full description of the underlying methods are detailed in reference[10], however for clarity, we summarise the key aspects relevant to the current analysis, and detail all additions and modifications to the original method, which were largely to account for the inclusion of ALOS-2 PALSAR-2 data covering the period 2015–2018.

**Cross calibration of ALOS and ALOS-2 mosaic product.** Systematic differences in backscatter values were observed between ALOS and ALOS-2, even in areas where the tree cover remained stable. The two sensors were cross-calibrated based on a regular grid (every 0.5 degrees) of pseudo-stable locations across the study region ($n = 1001$), excluding locations of possible forest change[33], steep slopes[53], and wetlands[54] Supplementary Fig. 7). A linear model ($\hat{\gamma}^0_{A2} = 1.001 \times \gamma^0_{A2} + 0.9478$; $r^2 = 0.71$; RMSE$_{dB} = 2.28$), was used to adjust the observed backscatter in dB from ALOS-2 ($\gamma^0_{A2}$) to that expected of ALOS ($\hat{\gamma}^0_{A2}$).

**Generating new composite maps of AGC.** For this study, we developed a new method to create datasets suitable for change detection using the freely available annual ALOS mosaic product, covering the ALOS measurement period from 2007 to 2010, and the first 4 years of ALOS-2 data from 2015 to 2018. A full

schematic diagram showing the rationale and procedure is shown in Figs. 6 and 7. The method was developed to account for spatial inconsistencies in the intensity of radar backscatter, which sometimes appear as clearly visible 'stripes' in the data between adjacent satellite acquisition paths (Fig. 6). The challenge for data users is these differences are often subtle and become apparent only when looking at the change between years. These patterns arise as L-band radar is sensitive to several factors unrelated to woody biomass, which in seasonally dry open woodlands and forests likely reflect differences in soil/vegetation moisture content at the time the data was acquired. The mosaic product comprises images obtained throughout the calendar year meaning there are some seasonal differences in when the data was acquired between neighbouring paths. In our previous study[10], a correction was applied to the radar data in areas where the estimated soil moisture differed between 2007 and 2010, and removed areas where the differences were too large to correct. Initial comparisons of the ALOS and ALOS-2 data, however, showed variations in soil moisture or precipitation could not fully account for observed discrepancies in backscatter. This likely reflects, in part, efforts to reduce or correct these anomalies by JAXA, which cannot easily be reversed, and may partly explain the weak relationship between backscatter, and both precipitation, and soil moisture.

The goal of this new method is to capture the general spatial and temporal trend in AGC stocks and land cover over the study period, while at the same time accounting for uncertainties associated with choosing a single, potentially biased pair of years (e.g. due to seasonal differences in the timing of the radar acquisition; Fig. 6). To do this, we remove from consideration any years, or parts thereof, which yield anomalously large positive or negative changes in backscatter, relative to surrounding areas, and to results obtained using different combinations of years. The data is aggregated from 25 m to 1 ha to reduce noise. The 2019–2021 mosaic products were excluded due to widespread striping issues which would likely have an undue effect on the derivation of a consistent trend, although updates to these datasets may allow their incorporation in future.

The method uses a spatial windowing technique, with the window size (240 km cross-track/East-West × 90 km along-track; ~North-South) chosen to include at least 3–4 acquisition paths for comparison, and reduce the likelihood that direct human change, i.e. widespread deforestation or degradation, has an effect on the results. For each iteration of the window, the relative change in backscatter for each pixel—expressed in annual terms (% yr$^{-1}$) given the range of years (5–11) covered—is calculated for all 16 bi-temporal combinations of data from ALOS (2007–2010) and ALOS-2 (2015–2018). The change datasets are then split based according to the combination of acquisition dates which make up the dataset to help pinpoint potential anomalies (Fig. 6). Three summary statistics describing the pattern of change across each of the unique path combinations are extracted; this includes the proportion of the wooded area (AGC density >10 Mg C ha$^{-1}$) that increased (or decreased) faster by the median rate of change across all 16 combinations of years to try and capture subtle discrepancies, or shifts in the data, irrespective of the intensity Fig. 7). The first and third quartiles of change are also extracted to account for areas where there are relatively large gains or decreases in backscatter e.g. flooded areas. An area is masked from that year combination if at least one of these change statistics lies outside the corresponding 10th and 90th percentiles, calculated using all pairwise combinations, i.e. areas that exhibit an abnormally high area and/or intensity of gain or loss relative to other years. The process is repeated with the centroid of each window shifting by one-third of the dimensions (i.e. 80 km W-E; 30 km N-S), with an area removed

**Rationale |** The annual mosaic contains inconsistencies and produces varying estimates of change depending on combination of years

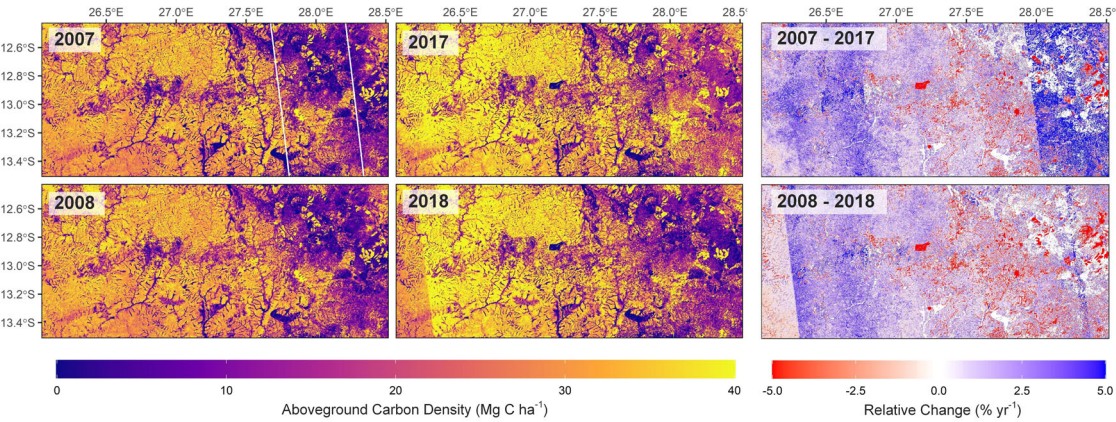

**Step 1 |** Create change maps for all possible year combinations, and split each map based on number of days between acquisition date

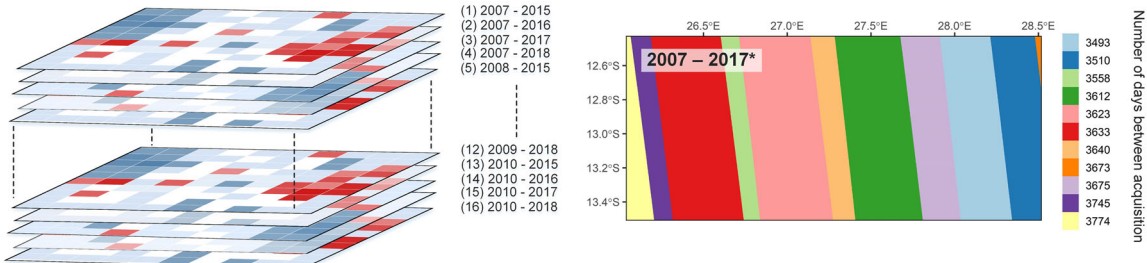

**Fig. 6 Generating composite maps of AGC for change detection (i).** The rationale behind the creation of the AGC maps used for change detection. The example data cover a single iteration of the moving window over a part of Central Zambia, chosen to highlight the potential ambiguity associated with choosing a single pair of years for analysis and where there is clearly visible 'striping' in data. The white lines in the AGC map for 2007 highlight a single pass of the ALOS satellite to show how the mosaic is constructed. The first step in the processing chain is to create change datasets for all possible combination of years, with these datasets further split based on the number of days between when the data was acquired (continues in Fig. 7).

from consideration if categorised as anomalous in more than half of all comparisons.

These masked datasets are then used to create a per pixel (25 m) weighted-average backscatter covering the ALOS years (2007–2010), and the ALOS2 (2015–2018) periods (Fig. 7), with lower, or no weight ascribed to years that yield potentially anomalous results. These composite, or harmonised datasets form the basis of our change analysis with the average time difference between the two being 8.2 years [IQR: 8–8.6]. The same pseudo-invariant areas used to create the cross-calibration model are used again to estimate the weighted standard error on the average backscatter estimates. The estimated uncertainty on each of the composite datasets is typically within 10% of mean across wooded areas, and decreases with increasing backscatter (Fig. 7f). This uncertainty is included in our probabilistic change detection method by simulating 10,000 possible values (realisations) of backscatter following a normal distribution, with the standard error parameter predicted for each possible backscatter value using 2nd order polynomial models fit to the data. In the original manuscript, per-pixel uncertainties were estimated using a model describing the effects of speckle, a noise-like quality inherent in SAR data, the effect of which is minimised by averaging over multiple years. For completeness, each simulated value of backscatter is then used to simulate 200 realisations of AGC, based on the standard error of the regression model using to convert backscatter to AGC, giving 2 million possible values of AGC for each possible observation of backscatter. The simulations are repeated for the full range of observed combinations of AGC at the first and second time points, and for each combination, the proportion of times that

the simulated values met the land cover change criteria is used the probability that the change has occurred. Uncertainties on all quantities were estimated through the propagation of the uncertainty in the model used to convert the radar data to estimates of AGC. This involves creating a set of 5000 different biomass-backscatter regressions using a different 50:50 split of the field plot data used to calibrate the model. These models are then applied to a random subsample of 5% the study area, each time calculating a new set of AGC and change estimates, with 2.5th and 97.5th percentiles of the 5000 estimates used as a measure of the uncertainty for the entire study area. A full description of our approach is contained in our previous manuscript[10].

**Land-cover change definitions**. Areas where AGC decreased over time are classified according to whether they are symptomatic of deforestation (a reduction in wooded area), degradation (a reduction in carbon density in an area that remains forest at both time points), or whether losses are of a lower intensity (<20% reduction in AGC), symptomatic of a minor, natural disturbance. Deforestation is defined as a reduction in AGC below a wooded/non-wooded threshold of 10 Mg C ha$^{-1}$, with degradation defined as a reduction in AGC density in an area that remains wooded. Both are separated from other disturbances according to whether the changes were of a high intensity, defined as a >20% reduction in AGC between time points. In this study, we included an additional separator, which is whether the absolute difference exceeded 5 Mg C ha$^{-1}$, or ~20% of the typical AGC density in these woodlands. This was done to prevent small

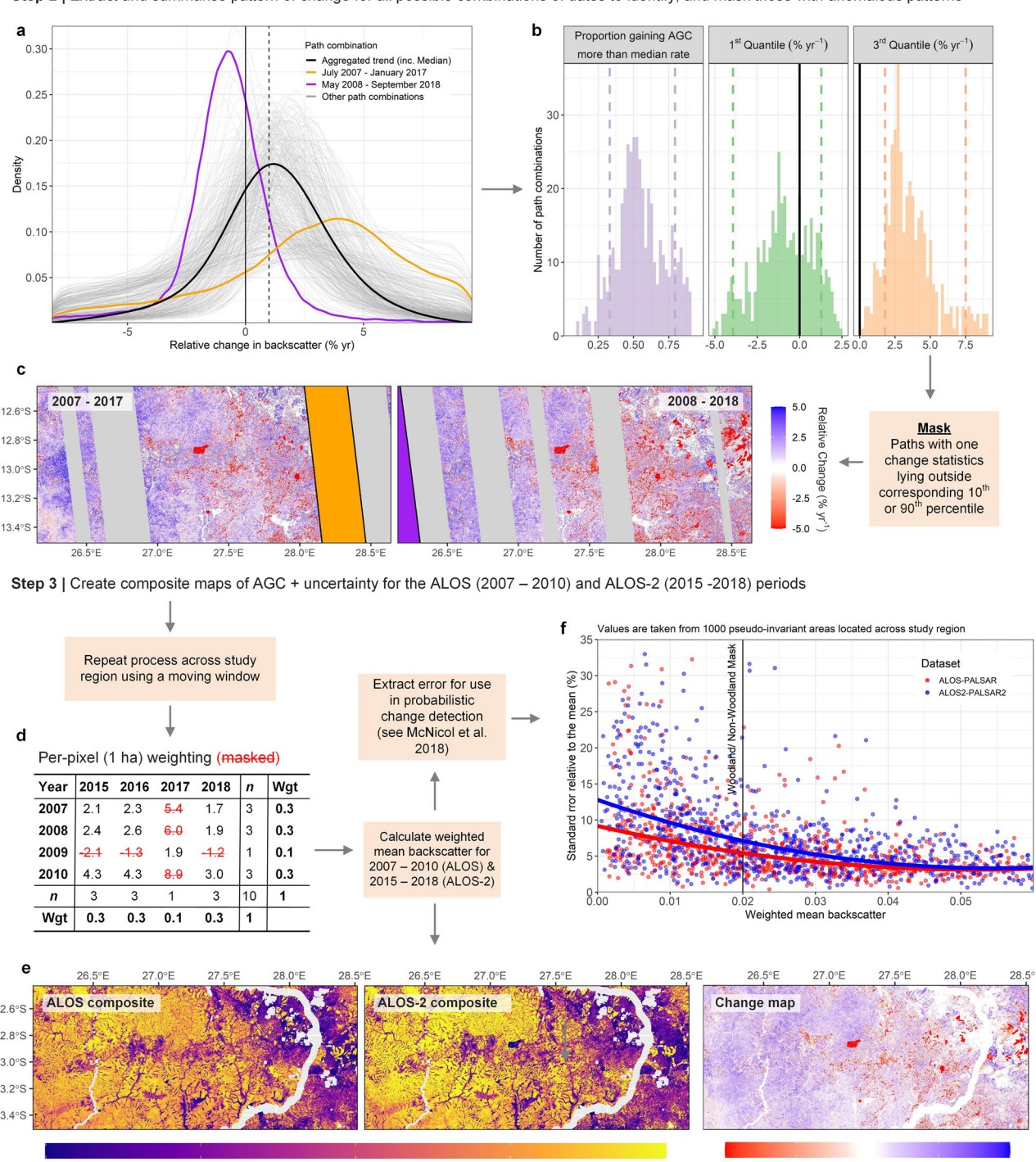

**Fig. 7 Generating composite maps of AGC for change detection (ii). a** The change data for each combination of satellite paths (based on number of days between acquisition) is extracted from across all possible combinations of years ($n = 16$), thus yielding hundreds of possible change scenarios for a given area. **b** These data is summarised using three change statistics which are used to identify and **c** mask areas with anomalous change patterns, i.e. areas which show a high area and/or intensity of gain or loss relative to surrounding areas, and other year combinations. This process is repeated across the study region, creating a set of masking layers for each year combination, which are then used **d–f** to calculate weighted mean backscatter and associated uncertainty for the ALOS (2007–2010) and part of the ALOS-2 period (2015–2018). The remainder of the processing follows ref. [10].

absolute, but large relative changes in areas with AGC densities close to the $10 \, \text{Mg C ha}^{-1}$ wooded/ non-wooded being designated as deforestation or degradation, when the size of the loss is more likely to be indicative of minor, natural disturbance. Our measure of deforestation represents clearances that are clearly visible and

semi-permanent, such as agriculture, whereas degradation will reflect more cryptic, patchy disturbances, such as logging or charcoal production[34,36]. Gains in AGC stocks reflect the growth of woody vegetation and are limited to areas that are wooded in the period 2007–2010.

**Protected areas**. The location of PAs were extracted from the UNEP/IUCN World Database on Protected Areas (WDPA) (May 2021)[55] with 1352 terrestrial PAs available for analysis within the political boundaries of the study region. There are 29 different designations describing PA strategy and management objectives, some of which are specific to each country. These were classified into three broad groups: (i) Strictly Protected, which includes any IUCN Category I–II areas, such as National Parks and Reserves, (ii) Forest Reserves, including both village and state managed reserves but excluding plantations, and (iii) Wildlife Reserves, which includes conservation and hunting areas. PAs designated after 2014 are excluded from the analysis as the impact of conservation unlikely to have been fully realised. At this point, PAs covered 260,000 km$^2$, or 18.7% of the land surface, with Tanzania and Zambia containing the largest networks with ~260,000 km$^2$ under protection, equivalent to 28% and 36% of the total land area respectively (Fig. 1). Forest Reserves are the most numerous protection type comprising 77% of all sites, although Wildlife Reserves and Strictly PAs are more spatially extensive constituting 43% and 36% of the entire PA network respectively.

**Matching**. In line with similar studies that have examined the ecological impact of protection in other tropical forest regions, we use matching methods to predict what would have happened to AGC and land cover in the absence of protection, i.e. the counterfactual outcome. The purpose of matching is to mimic the random assignment of protection by identifying unprotected areas that are similar to the PA of interest in terms of the probability of disturbance with the only difference being the assignation of protection. By comparing cells within each PA to similar unprotected areas, we aim to derive an unbiased estimate of the protection effect over the study period.

Here, we used coarsened exact matching (CEM)[29] to select protected and unprotected cells for comparison. In contrast to most other regional studies which tend to quantify the average performance across broad networks, matching was performed separately for each PA[37] with all the matched cells retained in order to calculate the effect size across networks of PAs, including those within the same country and of similar protection types. With CEM, the values for each matching covariate are first separated into discrete bins, the sizes of which are defined based on a priori knowledge of what is a suitable match; that is, where values in the same bin have little or no effect on the probability of the outcome (Supplementary Table 1). Exact matching is performed on these coarsened data, with any subsequent analyses on the matched dataset based on the original values. The effect size is calculated using a weighted difference in the means of the observed outcomes (i.e. the change in AGC, deforestation rate, degradation rate) in the protected and unprotected samples. In contrast to propensity score and multivariate matching, which require varying the calipers for each PA—i.e. the maximum acceptable difference between matched and control cells—both the bin sizes, and therefore the maximum degree of imbalance between matched cells is determined ex ante and are the same for all PAs, thus increasing transparency and ensuring differences in effect size are not due to differences in match quality.

**Matching covariates**. We controlled for a set of variables widely considered to affect the probability of the outcomes in question. These include the suitability of the area for agriculture, its extractable value and potential, and its accessibility, which includes geographic distance to markets and human populations. The rationale for each covariate, and the data sources and methods used to create these are in Supplementary Material. Control cells were located outside a 10 km buffer around all PAs[6,56] to avoid potential leakage effects influencing our results - i.e. the displacement of, or increase in activity outside the PA boundaries[57].

**Matching results**. The matching procedure greatly improved the comparability between the protected and unprotected sample (Supplementary Figs. 1 and 2). Of the 1346 individual PAs documented in the WDPA, 296 (22%) were excluded as they were too small to be considered for analysis (<1 km$^2$), while a further 152 (11%) were excluded as no suitable matches was found. However, the 904 PAs that were retained are amongst the largest in the region, encompassing the majority (98%) of the total protected wooded area, and of this, 56% received at least one match, with similar results (60 ± 31 [SD] %) when broken down by individual PA.

**Analysis**. The region-wide, and national results area calculated by pooling the matched data across each area. These results are presented according to the relative ($[\Delta_{PA} - \Delta_{UP}]/\Delta_{UP}$) change (%) in the outcome variables between the PA and its matched unprotected control sample due the differing size of each country, and thus magnitude of each change. For AGC change ($\Delta$AGC), positive values indicate that stocks in PAs are than unprotected areas, while for deforestation and degradation, a positive value indicates lower carbon emissions associated with that process. For the matching analysis, deforestation and degradation rates are primarily reported as a percentage of the land area, and not the wooded area unless otherwise stated. This is to prevent small differences in woody cover in the matched sample resulting in different rates of loss even when the total area affected is identical.

To examine the factors that affect PA effectiveness (Fig. 5 and Supplementary Fig. 6), we fitted a set of three Bayesian hierarchical regression models using the brms package[58] in R. The response variable in each case was the matched estimate of PA impact on deforestation, degradation or net AGC change and was modelled as arising from a Student's $t$ distribution whose parameters were estimated from the data. This response distribution was chosen to capture the fat-tailed nature of the response variables, in which a small number of PAs had very large estimated impacts. To account for the uncertainty in the matched estimates of PA impact we adopted a fixed meta-regression approach which also incorporated information about their associated standard errors. We used the size of the PA, average biomass density, distance from the nearest city, distance from other settlements, distance from the nearest road, population density in 2005, the roughness of the terrain, and indices of the suitability of the land for cash and subsistence farming, and a categorical variable classifying the management category of the PA as fixed predictor variables. We also included a random intercept for the country in which the PA is situated and allowed the effect of management type to vary by country. Both outcome and predictor variables were scaled and centred prior to modelling by subtracting the mean and dividing by two standard deviations. Results were subsequently back transformed to place them on the scale of the original data. Fixed effects were assigned weakly informative independent Normal priors with mean = zero and standard deviation = five and the random effects covariance's were assigned LKJ priors with regularisation parameters = two[59]. To facilitate interpretation of the fitted models, we predicted PA impact for all combinations of management type and country under scenarios in which all fixed predictors were held at their mean values. Our model-based results are presented as mean and their associated 95% and 80% CIs, and we interpret estimates whose 80% CIs do not overlap zero as statistically meaningful.

## Data availability

Data are available from the University of Edinburgh DataShare service at the following address: https://doi.org/10.7488/ds/7520.

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

## Acknowledgements

This work was initiated and partially funded by WWF-World Wide Fund for Nature on the project Dynamics of the Conservation Estate (DyCE) (Project Number: 10002150). I.M.M.'s time was also supported by the Natural Environment Research Council (NERC) and Department for International Development (DfID) funded Understanding the Impacts of the Current El Niño programme (NE/P004725/1) led by A.K., and a European Research Council Starting Grant awarded to E.T.A.M. for the Forest Degradation Experiment [FODEX] (757526). C.M.R.'s time was supported by the NERC funded SEOSAW (NE/P008755/1) and SECO projects (NE/T01279X/1). The radar data are freely available from JAXA, for which we are very grateful. We are also very thankful for the hard work of all those involved in the field data collection, acknowledgements for which can be found in our previous manuscript where this data was first used[10]. This work is a contribution to the Global Land Programme (https://glp.earth). We thank the two anonymous reviewers for their comments which improved the manuscript.

## Author contributions

I.M.M., A.K., N.D.B. and C.M.R. contributed to the conceptualisation and design of the study. I.M.M. processed and analysed the remote sensing data and other spatial datasets, developed the procedure for creating the composite AGC and AGC change maps, and performed the matching analysis. A.K. created the Bayesian regression models and S.J.B. developed the cross-calibration procedure for the radar data. All authors interpreted the results. I.M.M. led the writing of the manuscript with contributions from C.M.R. and A.K. S.J.B. and E.T.A.M. provided important comments. A.K., S.J.B., E.T.A.M. and C.M.R. provided funding. Correspondence and requests for materials should be addressed to I.M.M.

## Competing interests

The authors declare no competing interests.
