## [Peer Review File · Communications Earth & Environment]

26th Jun 23

Dear Dr McNicol,

Your manuscript titled "Protected areas reduce deforestation and degradation, and enhance woody growth across African woodlands" has now been seen by 2 reviewers, whose comments are appended below. You will see that they find your work of some potential interest. However, they have raised quite substantial concerns that must be addressed. In light of these comments, we cannot accept the manuscript for publication, but would be interested in considering a revised version that fully addresses these serious concerns.

We hope you will find the reviewers' comments useful as you decide how to proceed. Should additional work allow you to

- address these criticisms (that is, either to incorporate the suggestions or provide a compelling argument why the point made by the reviewer is not valid, or relevant to the editorial threshold as outlined below)

AND

- meet our editorial thresholds as outlined below,

then we would be happy to look at a substantially revised manuscript.

In the following, we list our main editorial concerns.

** Editorial threshold 1: provide compelling new insights into the effects of protected areas on changes in aboveground carbon stocks

** Editorial threshold 2: Discuss further the role of climate factors on the detected changes in aboveground biomass carbon stocks

** Editorial threshold 3: Demonstrate that the data and methods used to derive the changes are robust and accurately represent the effects of protected areas

If you choose to take up this option, please either highlight all changes in the manuscript text file, or provide a list of the changes to the manuscript with your responses to the reviewers.

If the revision process takes significantly longer than three months, we will be happy to reconsider your paper at a later date, as long as nothing similar has been accepted for publication at Communications Earth & Environment or published elsewhere in the meantime.

Please use the following link to submit your revised manuscript, point-by-point response to the reviewers' comments with a list of your changes to the manuscript text (which should be in a separate document to any cover letter), a tracked-changes version of the manuscript (as a PDF file) and any completed checklist:

[link redacted]

Please do not hesitate to contact us if you have any questions or would like to discuss the required revisions further. Thank you for the opportunity to review your work.

Best regards,

Jinfeng Chang, PhD
Editorial Board Member
Communications Earth & Environment
orcid.org/0000-0003-4463-7778

Alienor Lavergne, PhD
Associate Editor
Communications Earth & Environment

EDITORIAL POLICIES AND FORMAT

If you decide to resubmit your paper, please ensure that your manuscript complies with our editorial policies and complete and upload the checklist below as a Related Manuscript file type with the revised article:

Editorial Policy [Policy requirements](https://www.nature.com/documents/nr-editorial-policy-checklist.pdf) (Download the link to your computer as a PDF.)

For your information, you can find some guidance regarding format requirements summarized on the following checklist: (<https://www.nature.com/documents/commsj-phys-style-formatting-checklist-article.pdf>) and formatting guide (<https://www.nature.com/documents/commsj-phys-style-formatting-guide-accept.pdf>).

REVIEWER COMMENTS:

Reviewer #1 (Remarks to the Author):

This manuscript evaluates the effects of protected areas on changes in aboveground carbon. The topic is interesting and important, but I found the MS difficult to read and understand.

The AGC abbreviation appears multiple times but with different full names. For example, it was aboveground carbon stocks in the abstract (line 14) and aboveground carbon density in the introduction (line 74).

In Figure 1c, the y-axis label is carbon stock, with a unit of Pg C yr⁻¹ ? Moreover, the difference between 2007-2010 and 2015-2018 is hardly visible. This is also related to the bubble figures (e.g., Figure 2c), showing a magnitude of change below 1%, on which I have no clue how significant it was. Because this could be related to potential artifacts caused by cross-sensor differences between ALOS and ALOS2. The authors calibrated observations between the two sensors using a linear model, but the scatterplot in Extended Figure 7f shows a non-linear shift between them. How this will influence the estimated AGC?

ALOS composites were in 2007-2010 and 2015-2018. As I understood, they are mosaics every 4 years, but why they were separated again into individual years to calculate changes per year? This is very difficult to understand. I'm wondering how the estimated carbon stocks vary over the years, can you show some example time series?

Missing subplot label of 'c' in Figure 2.

Repeated subplot b in Figures 1 and 2.

lines 520-522: The matched unprotected areas are located within the 10km buffer zone, and the leakage effect of protected areas may be positive or negative. In other words, the establishment of protected areas may transfer human activities originally occurring inside protected areas to unprotected areas around protected areas. In this case, the forests within the protected areas were less disturbed and degraded, while the forest degradation in the adjacent areas around the protected areas was significantly increased (references below). However, it is difficult to uniform the leakage distance of different protected areas, which may require consideration of buffer zones of different distances or the dynamics of vegetation around the protected areas before and after the establishment of the protected areas.

Ewers RM, Rodrigues ASL (2008) Estimates of reserve effectiveness are confounded by leakage. *Trends in Ecology and Evolution*, 23, 113–116.

Oliveira PJ, Asner GP, Knapp DE, Almeyda A, GalvanGildemeister R, Keene S, Raybin RF, Smith RC (2007) Land-use allocation protects the Peruvian Amazon. *Science*, 317, 1233–1236.

Reviewer #4 (Remarks to the Author):

In this study, McNicol et al. study the effects of protected areas in African woodlands on aboveground carbon with respect to vegetation growth, deforestation, and degradation. Therefore, they created aboveground carbon maps from PALSAR/PALSAR2 data for two time periods. Then they studied carbon changes between those periods at different spatial scales ranging from the entire area to protected areas and matching areas outside the PAs. Further the effects of different socio-economic drivers were studied. The analyses show that protection has a positive effect on carbon,

but that protection status is less relevant. These results are highly relevant for developing management strategies these woody ecosystems and for policies required reduce greenhouse gas emissions. Given multiple threats for ecosystems including direct land use and climate change, such studies are highly relevant and timely, and provide strong support for the effectiveness of PAs.

Overall, the manuscript is well structured and well written. The analyses are robust and appropriate to answer the research questions. The authors developed a novel approach to derive harmonized carbon maps for the raw data; this approach is described in detail. The result support the main conclusions of the manuscript. The figures are of high quality.

I only have a few comments.

The analyses focus primarily on socio-economic drivers whereas climate drivers were not considered. I was wondering to which degree climate change or extreme events such as droughts or heat waves may have influence plant growth during the study period? While climate factors are not so relevant for degradation and deforestation, vegetation growth (and dieback) can be influenced by such factors. Similarly, climate factors were not included in the matching approach, climate variation is only mentioned as hidden bias. How similar are climate conditions in matching pairs of PA and non-PA? Are the responses in such pairs really comparable if climate conditions differ substantially? I suggest adding some discussion on the role of climate factors.

For the matching approach, the authors state that an advantage of the CEM approach is that the definition of thresholds for the acceptable difference between pairs is not required. However, binning variables also means that many thresholds are defined to define different bins and what is considered as a match regarding a variable. How sensitive are the results to the binning? Would changes in the thresholds for binning influence the matching and therefore the results?

Just out of interest: would it be possible to derive variables describing habitat structure from the remote sensing data? For example, vegetation height, density of tall and small trees, herbaceous cover etc. Such variable might be highly relevant for management in PAs, maybe even more relevant than carbon. For example, game reserves might be more interested in open vegetation and a habitat structure suitable for wildlife than in carbon sequestration. Similar to the analyses presented for carbon, the effects of protection status on habitat structure could be analyzed. (I'm not suggesting to include this in this study.)

To which degree can growth be attributed to afforestation or plantations? Did such activities take place in the study region, particularly in regions where carbon increased?

In Fig 2 and 3 (and similar figures) some of the numbers in the circles are very close to each other or even overlap. I suggest to modify the figures such that all numbers can be read, for example by writing the numbers next to the circles.

I think some sentences are quite long, for example lines 45-50. Please consider to split.

L 76: the world's largest savanna ecosystem – can this entire study region be considered as one ecosystem?

L 290 “degradation accounted for ... (ie deforestation and degradation)...” so only degradation or

both? Please clarify.

L 360 I suggest to put an important conclusion at the end of the manuscript, such as implications for management, not a more or less technical point.

Author responses to reviewer comments on “Protected areas reduce deforestation and degradation, and enhance woody growth across African woodlands” by McNicol et al.

We thank the editors and the reviewers for their positive and helpful comments which have improved the manuscript. As always, we appreciate the time and effort that goes into the review process. We have endeavoured to address all of the comments in detail, and we hope that you now consider our manuscript ready for publication.

Please note that all edits to the text are highlighted yellow in the revised manuscript.

Kind regards,

Dr. Iain McNicol on behalf of all co-authors

Reviewer #1 (Remarks to the Author):

[1] This manuscript evaluates the effects of protected areas on changes in aboveground carbon. The topic is interesting and important, but I found the MS difficult to read and understand.

Response:

We thank the reviewer for taking the time to review this manuscript, and for their attention to detail on our figures and methods.

—

[2] The AGC abbreviation appears multiple times but with different full names. For example, it was aboveground carbon stocks in the abstract (line 14) and aboveground carbon density in the introduction (line 74).

Response:

Well spotted. We have changed the text so the AGC abbreviation refers solely to “aboveground woody carbon”, e.g. in the abstract, the text now reads “aboveground woody carbon (AGC) stocks” as opposed to “aboveground woody carbon stocks (AGC)”.

—

[3] In Figure 1c, the y-axis label is carbon stock, with a unit of Pg C yr⁻¹?

Response:

Again, well spotted. This has been changed to Pg C.

[4] ALOS composites were in 2007-2010 and 2015-2018. As I understood, they are mosaics every 4 years, but why were they separated again into individual years to calculate changes per year? This is very difficult to understand. I'm wondering how the estimated carbon stocks vary over the years, can you show some example time series?

Response:

Yes, we map AGC at two time-points, with one map covering the period 2007 - 2010, and another covering the period 2015 - 2018. These are created by merging the annual mosaic datasets for these individual years, excluding those years, or parts thereof, that our method identified as producing anomalous results. We do not measure changes on an annual basis and/or as part of a time-series analysis, rather we measure the difference in AGC between these two composite/ harmonized maps.

We believe the confusion here is because changes between these two datasets are expressed on an annual time-scale. The reason for this is the average time-difference between the two datasets varies depending on which years in the ALOS and ALOS2 periods were considered suitable for inclusion. For example, for an area where only data from 2007 and 2008 were included for the ALOS period, and 2017 - 2018 for the ALOS-2 period, the average time difference in that part of the change map would be around 10 years (2017.5 - 2007.5).

Overall, the average time-difference between the two composite maps is 8.2 years [interquartile range: 8 - 8.6 years], though it can be as low as 5 - 6 years (e.g. 2009/10 - 2015/16, and as high as 11 (i.e. 2007 - 2018). If we did not account for this then our results would be subject to bias as we would allow more change to occur in some areas, and less in others.

We have included an abridged version of the above paragraph on L451, and included similar text in the legend for Figure 2:

"Changes are expressed on an annual basis as the average time-difference between the two maps varies across the study region"

[5] [i] Moreover, the difference between 2007-2010 and 2015-2018 is hardly visible. This is also related to the bubble figures (e.g., Figure 2c), showing a magnitude of change below 1%, on which I have no clue how significant it was.

[ii] Because this could be related to potential artifacts caused by cross-sensor differences between ALOS and ALOS2. The authors calibrated observations between the two sensors using a linear model, but the scatterplot in Extended Figure 7f shows a non-linear shift between them. How will this influence the estimated AGC?

Response:

A very good question concerning the uncertainties on our estimates.

In response to the first part of the comment [i], we do calculate the uncertainties on these changes, however we realise these were not stated for all reported quantities when first mentioned in the main text, including the relative change (% yr⁻¹), or the matched estimates. We worried about reducing interpretability of the figures, but think we got the balance wrong and the paper is clearer with these uncertainties included in the main text and figures. We have therefore edited the main text and figures accordingly to include uncertainty bounds (Figure R1).

Figure R1 - Updated Figure 2c including the 95% confidence bounds

As an example of how we've changed the text, we now state that on L165 – 168 in reference to the Figure 2c that “we find PAs have an overall positive effect on aboveground woody carbon (AGC), with stocks increasing by +0.53% yr⁻¹ [0.43 – 0.62% yr⁻¹] (2.8 [2.2 – 3.5] Tg C yr⁻¹) inside matched PAs, compared to +0.08% yr⁻¹ [-0.05 – 0.21% yr⁻¹] (0.40 [-0.23 – 1.12] Tg C yr⁻¹) in the matched unprotected sample (Fig. 2)”.

The procedure for how we calculate these uncertainties is now described in more detail on lines 455 – 474, and a full explanation is contained in the Supplementary Materials of our previous manuscript (McNicol et al., 2018). However, in short, we carefully account for, and propagate the uncertainties associated with the harmonised maps (the data presented in Figure 7f - see paragraph below), and the uncertainties on the linear model used to convert the radar data into AGC. This involves re-calculating all of the reported quantities using 5000 different biomass-backscatter regressions based on a 50:50 split of the field plot data used to calibrate the model.

From these 5000 estimates, we extract the 2.5th and 97.5th percentiles as a measure of the uncertainty on our estimates. In practice, this leads to 95% CIs for the change in AGC between the two time points that range from 93% to 109% of the reported 'best-guess' estimate. In situations where these confidence bounds overlap with zero, one could interpret the change as not being meaningfully different from zero. However, this is not the case where the uncertainty bounds on the protected, and unprotected estimates overlap as an iteration of the biomass-backscatter model which yields a lower estimate of change inside a protected area would also yield a lower estimate inside an unprotected area.

The data presented in Figure 7f show the variation in backscatter across the individual years that make up each of the harmonized maps, compared to the averaged value. There is a non-linear trend because radar backscatter is more sensitive to surface moisture in more open areas and will vary more between annual mosaics if the data in these is acquired at very different times of the year. We now realise this sub-figure was not fully explained in the legend, nor was it specifically referenced in the main text alongside the explanation of this part of methods, both of which have been rectified (L453 - 455).

[ii] Because this could be related to potential artifacts caused by cross-sensor differences between ALOS and ALOS2. The authors calibrated observations between the two sensors using a linear model, but the scatterplot in Extended Figure 7f shows a non-linear shift between them. How will this influence the estimated AGC?

With regards to the second part of the comment [ii], we do indeed use a linear model to correct for the cross-sensor differences between ALOS and ALOS2 (Figure R2). We do this using data collected from 1000 sites which are assumed not to have experienced significant change over the study period. This is normal practice when doing comparisons between sensors which have not otherwise been normalized, for example used for NDVI between different Landsat satellites (Janssen et al., 2018).

It is well known that ALOS-2 PALSAR-2 is a higher powered sensor than ALOS PALSAR, and therefore produces higher backscatter values for the same amount of woody biomass (see slide 15 in the link in footnote¹). The relationship that we found between ALOS PALSAR and ALOS-2 PALSAR-2, as we expected, showed an upwards bias in the ALOS-2 data relative to the ALOS data (Figure R2). If we had not done this cross-calibration process to account for this increase in power, this would have led to the false detection of biomass growth in areas where no such change has occurred, and would miss large areas of deforestation and degradation. Instead, we find that total AGC stocks for the study region are not increasing rapidly, but are broadly in balance, which as a sanity check, is broadly in keeping with our previous findings.

We hope that this addresses the reviewer's questions.

¹https://eo4society.esa.int/wp-content/uploads/2021/01/2015_3rdPolarimetry_PALSAR-2_MShimada.pdf

Figure R2 - The ALOS2 - ALOS cross calibration model based on data collected from 1000 sites where no significant change is likely to have occurred over the study period. The model is used to bring ALOS-2 in line with the ALOS data. This Figure has now been included in the Supplementary Material.

[6] Missing subplot label of 'c' in Figure 2.
Repeated subplot b in Figures 1 and 2.

Added the subplot label c to the figure. We kept the same subplot b in both Figures 1 and 2 so the figures could stand alone, and the reader could better compare the patterns in AGC storage (Figure 1), and AGC change (Figure 2) to the location of PAs. We have added a small piece of text to the legend of Figure 2 to explain this reasoning:

“(b) The location of protected areas in 2014 separated by broad management type is included again to ease comparisons between patterns of AGC change, and PA location”

[7] lines 520-522: The matched unprotected areas are located within the 10 km buffer zone, and the leakage effect of protected areas may be positive or negative. In other words, the establishment of protected areas may transfer human activities originally occurring inside protected areas to unprotected areas around protected areas. In this case, the forests within the protected areas were less disturbed and degraded, while the forest degradation in the adjacent areas around the protected areas was significantly increased (references below).

However, it is difficult to uniform the leakage distance of different protected areas, which may require consideration of buffer zones of different distances or the dynamics of vegetation around the protected areas before and after the establishment of the protected areas.

Ewers RM, Rodrigues ASL (2008) Estimates of reserve effectiveness are confounded by leakage. *Trends in Ecology and Evolution*, 23, 113–116.

Oliveira PJ, Asner GP, Knapp DE, Almeyda A, Galvan, Gildemeister R, Keene S, Raybin RF, Smith RC (2007) Land-use allocation protects the Peruvian Amazon. *Science*, 317, 1233–1236.

Response:

This seems to be a confusion. On L541 - 543 (in new draft) we state “*Control cells were located outside a 10 km buffer around all protected areas to avoid potential leakage effects influencing our results - i.e. the displacement of, or increase in activity outside the PA boundaries (Ewers and Rodrigues, 2008)*”. This was done for the reasons described - to reduce the potential impact of human activities which may have occurred inside protected areas being transferred to unprotected areas around protected areas.

The second part of the comment, which we understand to be the key part, is the rationale behind the use of a 10 km buffer around PAs, and whether buffer zones of different size(s) should have been considered based on local circumstances. It is true that the extent of local spillover may vary across the study region depending on level of threat, socio-economic conditions, and the influence of tourism (e.g. Robalino et al., 2017 - citation included in main text). A 10 km buffer is common in the literature, and appears to be based on earlier work by Andam et al. (2008) in Costa Rica. The aforementioned paper by Robalino shows that this buffer was appropriate in Costa Rica and was also considered appropriate here given that most deforestation and degradation is to satisfy local demand for resources, often occurring with 3 - 5 km of settlements or roads (see citations in Supplementary Table 1). Thus, in a situation where settlements or roads grow up close to PA boundaries, the effect is unlikely to extend much beyond this distance.

Reviewer #4 (Remarks to the Author):

[8] In this study, McNicol et al. study the effects of protected areas in African woodlands on aboveground carbon with respect to vegetation growth, deforestation, and degradation. Therefore, they created aboveground carbon maps from PALSAR/PALSAR2 data for two time periods. Then they studied carbon changes between those periods at different spatial scales ranging from the entire area to protected areas and matching areas outside the PAs. Further the effects of different socio-economic drivers were studied. The analyses show that protection has a positive effect on carbon, but that protection status is less relevant. These results are highly relevant for developing management strategies for these woody ecosystems and for policies required to reduce greenhouse gas emissions. Given multiple threats for ecosystems including direct land use and climate change, such studies are highly relevant and timely, and provide strong support for the effectiveness of PAs.

Overall, the manuscript is well structured and well written. The analyses are robust and appropriate to answer the research questions. The authors developed a novel approach to derive harmonized carbon maps for the raw data; this approach is described in detail. The results support the main conclusions of the manuscript. The figures are of high quality.

Response:

We thank the reviewer for taking the time to review this manuscript, and for their insightful comments on our methods and results.

—

[9] [i] The analyses focus primarily on socio-economic drivers whereas climate drivers were not considered. I was wondering to which degree climate change or extreme events such as droughts or heat waves may have influenced plant growth during the study period? While climate factors are not so relevant for degradation and deforestation, vegetation growth (and dieback) can be influenced by such factors.

[ii] Similarly, climate factors were not included in the matching approach, climate variation is only mentioned as hidden bias. How similar are climate conditions in matching pairs of PA and non-PA? Are the responses in such pairs really comparable if climate conditions differ substantially? I suggest adding some discussion on the role of climate factors.

Response:

This is a very good question about the potential for climatic differences between areas to have an effect on our estimates of vegetation growth.

With regards to the matching analysis [ii], climate conditions could have an effect on the results if PA and non-PA samples were selected from very different geographic areas. However, we

selected matched control (i.e. non-PA) samples from the same administrative regions as the PA in question to account for this very thing, and other potentially hidden biases. We also matched areas using modeled estimates of agricultural suitability (FAO and IIASA, n.d.), which includes climate information in its predictions. Our 'relative elevation' metric was also considered a proxy for temperature differences, all of which taken together was assumed to prevent matched PAs and non-PAs being located in areas with markedly different climatic conditions.

That being said, we tested these assumptions by comparing rainfall and temperature patterns in the matched samples using monthly estimates from WorldClim (<https://www.worldclim.org/data/monthlywth.html>) covering the period 2010 - 2019 (last year excluded to fit with study period). This includes comparisons of the general climate conditions, including the mean annual total precipitation (mm/ year) and mean seasonality - the variation of rainfall within a single year, expressed by the coefficient of variation - and the mean annual maximum temperature. To capture extremes, we also extracted the 10th and 90th percentiles of the monthly precipitation data across all years (n months = 108) as a proxy for drought and flooding respectively, and the 90th percentile of the monthly maximum temperature data as a proxy for heatwaves.

Figure R3 - Post-hoc comparison of the climate conditions in protected and unprotected areas before and after matching. Quantile-Quantile plots are used to compare the climate conditions in the pre- and post-matching samples. If the points lie along the 1:1 line then both the protected, and unprotected data have a similar distribution. If the points lie below the 1:1 line, then a larger proportion of PAs have those values than in unprotected areas.

The results of this show no meaningful difference in these variables between matched PAs and non-PAs, meaning our estimates of PA effectiveness are robust to climatic conditions (Figure R3). We have included these figures in the Supplementary Material (Supplementary Figure 2), and added a section in our description of the matching covariates which copies the above paragraph near verbatim.

With regards to the first point [i] as to whether climate differences are likely to have influenced plant growth, the results also show that even prior to matching, protected and unprotected areas do not exhibit clear differences in climate. The exception to this is the 10th percentile of precipitation which we use as a proxy for drought and/or dry season length, which we find is more prevalent/ longer in protected areas, although the difference is small.

We have also included an acknowledgment on L279 that areas of vegetation growth also appear to be co-located with areas where MAP is relatively high:

“Our results reaffirm known patterns of AGC change across our study region¹⁰ [reference is to our previous manuscript], with carbon losses concentrated around urban centres and road networks, and carbon gains in more remote areas with sparse population, and/or a relatively high (> 1000 mm yr⁻¹) mean annual precipitation (e.g. northern Angola, western Zambia, southern DRC).”

The question of what is driving the patterns of loss (e.g. commercial vs shifting agriculture, logging) and growth (e.g. differences in rainfall, increasing CO₂) is certainly a topic worth exploring, however we feel this should be as part of a companion study to this analysis which is focused on the role of protected areas.

[10] For the matching approach, the authors state that an advantage of the CEM approach is that the definition of thresholds for the acceptable difference between pairs is not required. However, binning variables also means that many thresholds are defined to define different bins and what is considered as a match regarding a variable. How sensitive are the results to the binning? Would changes in the thresholds for binning influence the matching and therefore the results?

Response:

The decision of what constitutes an appropriate comparator is a central part of all matching analysis. The stated advantage of the CEM method is that compared to other methods like propensity score matching, the thresholds used to group covariates for comparison, i.e. the maximum level of imbalance between groups of PA and non-PA samples, are transparent, and the same across all PAs, reducing the chance that differences in effect scores are caused by differences in match quality.

In the Supplementary Information, we lay out in detail the thresholds and why differences within these bins are considered acceptable for comparison, which is a necessary step for any study using CEM. In Figure S1, we show that we achieve a very good balance between the PA and non-PA samples, which is the key outcome for any matching method, and in our previous comment, show that our sample set is not affected by differences in climate conditions.

However, as an additional sanity check, we re-performed the matching analysis a further two times using slightly narrower thresholds for certain layers where we feel such changes would still be permissible.

This includes:

- 1) Changing the bins for the ‘travel time to nearest city’ layer from 120 minute bins if > 3 hours away, to 60 min minute bins throughout, and adding an extra bin into the population density layers between 100 and 250 people per km² to further separate areas around larger cities.
- 2) Retaining the previous changes, but this time adding an additional bin to the population density layers at 25 people per km² to further separate very remote areas with low population densities from smaller rural areas. We also narrowed the thresholds used in the ‘topographic roughness’ layer (i.e. standard deviation of 90 m elevation within a 5 x 5 pixel grid) to bins of 10 m, rather than 15 m.

We find that varying these variables has a very small impact on the results (Table R2), and does not change the overall conclusions that protected areas reduce deforestation and degradation, and enhance biomass growth.

Table R2 - Comparison of key change statistics as derived using different binning thresholds. The column “*n*” is the total area of protected lands (km²) that were matched in each iteration

Variables	Original (n = 283,480)		Re-analysis (1) (n = 250,204)		Re-analysis (2) (n = 224,349)	
	PA	non-PA	PA	non-PA	PA	non-PA
AGC Change (% yr ⁻¹)	0.53	0.07	0.50	0.06	0.52	0.09
Deforestation rate (% yr ⁻¹)	0.39	0.64	0.41	0.65	0.39	0.63
Degradation rate (% yr ⁻¹)	0.66	0.86	0.66	0.87	0.65	0.84
Vegetation growth (% yr ⁻¹)	6.18	5.53	6.13	5.5	6.12	5.51

[11] Just out of interest: would it be possible to derive variables describing habitat structure from the remote sensing data? For example, vegetation height, density of tall and small trees, herbaceous cover etc. Such variables might be highly relevant for management in PAs, maybe even more relevant than carbon. For example, game reserves might be more interested in open vegetation and a habitat structure suitable for wildlife than in carbon sequestration. Similar to the analyses presented for carbon, the effects of protection status on habitat structure could be analyzed. (I'm not suggesting to include this in this study.)

Response:

Yes it is possible to measure these variables, but not with the type of satellite data used in this study. We made a point of noting in the discussion that carbon sequestration may not be the core aim of protection, such as in game reserves. However, our results do provide information on the level of human activity/ disturbance in these areas which may be of interest to practitioners working in these areas.

There are several ways one could estimate habitat structure, one of which would be to use Interferometric Synthetic Aperture Radar (InSAR) which uses information collected from multiple overlapping radar images to get three-dimensional information of the earth surface, such as vegetation height. This can also be monitored over time to look for disturbances.

Another way to do this would be to use spaceborne LiDAR instruments, such as GEDI and ICESat-2, to estimate 3D tree structure. The drawback of these instruments is they sample a relatively small part of the surface, meaning that other satellite datasets are needed to extrapolate these measurements to create the “wall-to-wall” datasets needed for mapping across large PA networks.

In more open areas, herbaceous cover could be mapped at relatively high resolution using optical images from Sentinel-2, and Landsat, the latter of which has a long temporal record (decades) which would be useful for long term monitoring.

Doing any of these studies alone would be a paper itself, though combining these to measure changes in more open savanna areas would be a very interesting study.

—

[12] To which degree can growth be attributed to afforestation or plantations? Did such activities take place in the study region, particularly in regions where carbon increased?

Response:

A good question. We checked this using data from Lesiv et al (2022) which maps forest management status across the globe at 100 m resolution and includes the location of planted and plantation forests. From this we estimated that a very small percentage of the woody area in each country are areas classified as plantation (Table R1).

Table R1 - Comparison of the estimated plantation area (ha) in areas we classified as being wooded lands, compared to the total woody area for each country (1000s ha).

Country	Angola	Katanga	Malawi	Mozambique	Tanzania	Zambia	Zimbabwe
Woody area (1000s ha)	67,868	31,732	3,404	49,552	45,505	41,912	13,984
Plantation Area (ha)	9,054	5,262	22,624	13,842	72,439	16,103	91,975
Plantation as % of woody area	0.01	0.16	0.66	0.02	0.16	0.04	0.44

The World Database of Protected Areas (WDPA) includes 58,000 of forest plantations in Tanzania which were removed from the analysis. In the interests of transparency, we found a recent study by Andrew (2022) which suggests there is 600,000 ha of planted/ plantation forest in Tanzania based on government statistics, although the majority of this (400k ha) is thought to be small to medium growers (size undefined). Even if we were to take this value as an upper limit and assume all of this area was detected as being 'wooded' at the start of the study, then planted forests would still only comprise a 1.3% of dataset for Tanzania. Another recent study by Bey and Meyfroidt (2021) showed that in Northern Mozambique (35% of the total country area), only 0.5% of the land area was covered by plantations.

To summarise, based on the data available to us it is very unlikely that plantations will have a meaningful impact on our estimates of deforestation and growth. Our maps also show that rather than being concentrated in specific areas, vegetation growth is widespread across the region, particularly in areas far from urban centers and road networks (e.g. north-eastern Angola).

—
 [13] In Fig 2 and 3 (and similar figures) some of the numbers in the circles are very close to each other or even overlap. I suggest to modify the figures such that all numbers can be read, for example by writing the numbers next to the circles.

Response:

The figures have been modified to make sure the numbers are clearly separated. We also note that the scale used in Figure 3 has also changed from being a relative change, e.g. deforestation losses relative to the initial AGC stock in each area, to the average total (gross) loss or gain within a 1km² grid cell. This was done to so that the values presented were more in keeping with the values reported in the text.

[14] I think some sentences are quite long, for example lines 45-50. Please consider to split.

Response:

Agreed. The cited sentence has been split up, as have others which we felt upon re-reading were too long.

[15] L 76: the world's largest savanna ecosystem – can this entire study region be considered as one ecosystem?

Response:

This is a good point. The dominant vegetation types are savanna (miombo) woodlands, however, the region does contain a variety of vegetation types including coastal and montane forests.

We have edited the text to say: *“Here, we present new datasets showing changes in aboveground woody carbon (AGC), and associated changes in land cover, including deforestation, degradation and growth across the world’s largest savanna ecoregion – the southern African woodlands – which cover 2.5 million km² over 7 countries (Fig. 1).”*, which is more appropriate and in keeping with previous descriptions of the region (Timberlake and Chidumayo, 2011).

[16] L 290 “degradation accounted for ... (ie deforestation and degradation)...” so only degradation or both? Please clarify.

Response:

We have modified the text to hopefully make this clearer:

*“In this study, we find that degradation accounted for half (49%) of the **combined** anthropogenic (i.e. deforestation and degradation) carbon losses, confirming its importance as a major source of land use change emissions.”*

[17] L 360 I suggest to put an important conclusion at the end of the manuscript, such as implications for management, not a more or less technical point.

Response:

We added the following italicized text to the end of the manuscript:

“However, the advantages of the longer time frame is that it leads to higher signal to noise ratios for detecting land cover change. *As such, we can be more confident that the observed changes are real and that our findings accurately reflect the impact of protected areas on the regional carbon balance.*

Citations

- Andrew, S.M., 2022. Drivers, trends and management of forest plantation fires in Tanzania. *Trees For. People* 10, 100355. <https://doi.org/10.1016/j.tfp.2022.100355>
- Bey, A., Meyfroidt, P., 2021. Improved land monitoring to assess large-scale tree plantation expansion and trajectories in Northern Mozambique. *Environ. Res. Commun.* 3, 115009. <https://doi.org/10.1088/2515-7620/ac26ab>
- Dons, K., Panduro, T.E., Bhattarai, S., Smith-hall, C., 2014. Spatial patterns of subsistence extraction of forest products - an indirect approach for estimation of forest degradation in dry forest. *Appl. Geogr.* 55, 292–299. <https://doi.org/10.1016/j.apgeog.2014.08.018>
- FAO and IIASA, n.d. Global Agro-Ecological Zones (GAEZ v3).
- Green, J.M.H., Larrosa, C., Burgess, N.D., Balmford, A., Johnston, A., Mbilinyi, B.P., Platts, P.J., Coad, L., 2013. Deforestation in an African biodiversity hotspot: Extent, variation and the effectiveness of protected areas. *Biol. Conserv.* 164, 62–72. <https://doi.org/10.1016/j.biocon.2013.04.016>
- Janssen, T.A.J., Ametsitsi, G.K.D., Collins, M., Adu-Bredu, S., Oliveras, I., Mitchard, E.T.A., Veenendaal, E.M., 2018. Extending the baseline of tropical dry forest loss in Ghana (1984–2015) reveals drivers of major deforestation inside a protected area. *Biol. Conserv.* 218, 163–172. <https://doi.org/10.1016/j.biocon.2017.12.004>
- McNicol, I.M., Ryan, C.M., Mitchard, E.T.A., 2018. Carbon losses from deforestation and widespread degradation offset by extensive growth in African woodlands. *Nat. Commun.* 1–19. <https://doi.org/10.1038/s41467-018-05386-z>
- Robalino, J., Pfaff, A., Villalobos, L., 2017. Heterogeneous Local Spillovers from Protected Areas in Costa Rica. *J. Assoc. Environ. Resour. Econ.* 4, 795–820. <https://doi.org/10.1086/692089>
- Timberlake, J., Chidumayo, E., 2011. Miombo Ecoregion Vision report, Occasional Publications in Biodiversity. Biodiversity Foundation for Africa, Bulawayo, Zimbabwe.
- Wessels, K.J., Colgan, M.S., Erasmus, B.F.N., Asner, G.P., Twine, W.C., Mathieu, R., van Aardt, J. a N., Fisher, J.T., Smit, I.P.J., 2013. Unsustainable fuelwood extraction from South African savannas. *Environ. Res. Lett.* 8, 014007. <https://doi.org/10.1088/1748-9326/8/1/014007>

12th Sep 23

Dear Dr McNicol,

Your revised manuscript titled "Protected areas reduce deforestation and degradation, and enhance woody growth across African woodlands" has now been seen by our reviewers, whose comments appear below. In light of their advice we are delighted to say that we are happy, in principle, to publish a suitably revised version in Communications Earth & Environment under the open access CC BY license (Creative Commons Attribution v4.0 International License).

We therefore invite you to revise your paper one last time.

Please consider incorporating some of your responses to the reviewers into the main text and methods section of the manuscript, to provide context for readers regarding your methods and results.

At the same time we ask that you edit your manuscript to comply with our format requirements and to maximise the accessibility and therefore the impact of your work.

EDITORIAL REQUESTS:

*****Please take care to match our formatting and policy requirements. We will check revised manuscript and return manuscripts that do not comply. Such requests will lead to delays. *****

SUBMISSION INFORMATION:

OPEN ACCESS:

Communications Earth & Environment is a fully open access journal. Articles are made freely accessible on publication under a [CC BY license](http://creativecommons.org/licenses/by/4.0) (Creative Commons Attribution 4.0 International License). This license allows maximum dissemination and re-use of open access materials and is preferred by many research funding bodies.

For further information about article processing charges, open access funding, and advice and support from Nature Research, please visit <https://www.nature.com/commsenv/article-processing-charges>

At acceptance, you will be provided with instructions for completing this CC BY license on behalf of all authors. This grants us the necessary permissions to publish your paper. Additionally, you will be asked to declare that all required third party permissions have been obtained, and to provide billing information in order to pay the article-processing charge (APC).

[link redacted]

Best regards,

Heike Langenberg, PhD
Chief Editor
Communications Earth & Environment

Jinfeng Chang, PhD
Editorial Board Member
Communications Earth & Environment
orcid.org/0000-0003-4463-7778

On Twitter: @CommsEarth

REVIEWERS' COMMENTS:

Reviewer #1 (Remarks to the Author):

I would like to thank the authors for making necessary revisions and responses to my comments. I think the paper is now ready for publication.

Reviewer #4 (Remarks to the Author):

All comments and suggestions of the review were considered and I don't have any further comment.

Reviewer comments on “Protected areas reduce deforestation and degradation, and enhance woody growth across African woodlands” by McNicol et al.

The reviewers provided no additional questions or suggestions on the manuscript.

Reviewer #1 (Remarks to the Author):

[1] I would like to thank the authors for making necessary revisions and responses to my comments. I think the paper is now ready for publication.

Reviewer #4 (Remarks to the Author):

[2] All comments and suggestions of the review were considered and I don't have any further comment.